# Defensive Strongholds and Fortified Castles in Poland—From the Art of Fortifications to Tourist Attractions

Jacek Kulpiński *, Beata Prukop, Paweł Rut, Aneta Rejman, Paweł Świder and Wojciech J. Cynarski

Institute of Physical Culture Studies, College of Medical Sciences, University of Rzeszow, 35-959 Rzeszów, Poland; bprukop@ur.edu.pl (B.P.); prut@ur.edu.pl (P.R.); anetarejman13@wp.pl (A.R.); pswider@ur.edu.pl (P.Ś.); ela_cyn@wp.pl (W.J.C.)
* Correspondence: qlax@wp.pl

**Abstract:** The scientific problem undertaken is the importance of castles for the functioning of cultural tourism in the opinion of the inhabitants of Central Europe. What is the use of medieval monuments for the art of fortifications today? The main method of research is a diagnostic survey carried out with the use of a survey on a group of N, important according to the statistics in the number of $n = 614$ respondents. Statistical analyzes were performed using Statistica version 13.3. On the basis of the presented research results, it can be concluded that the interests of the respondents are very broad and varied. Taking into account the relatively large group of respondents, the research results can be considered reliable. An important goal was supplementing the knowledge, meeting the needs of learning about history, and getting acquainted with the prevailing historical tradition in castles of Europe. The questions presented here accent the interests of castles for the functioning of cultural tourism in the opinion of the inhabitants of Central Europe.

**Keywords:** tourism theory; cultural tourism; martial arts; defensive stronghold; castle

## 1. Introduction

Reflections on the art of fortifications concern in this case especially the Middle Ages and the strongholds and fortified castles built at that time. Their genesis in Poland makes us go back to the Bronze Age and the so-called Lusatian culture. The art of fortification, the manifestation of which are castles that have survived to this day, is one of the forms of the art of war. As part of the General Theory of Fighting Arts (GTFA), not only are traditional weapons and technical and tactical issues of combat taken into account but also building fortifications and the art of obtaining them. In their history, Poles became famous not only for the perfect training of a warrior, knight and soldier, horse fighting, unmatched fencing and traditional bravery in combat, but they are also the creators of original achievements in the field of the art of fortifications. The achievements of former Poles and the passions of their contemporary followers (reconstruction groups, people cultivating old Polish sword fencing, etc.) are in the area of interest of martial arts researchers from the GTFA perspective [1,2].

The theory of tourism is useful not only for practising tourism, organizing tourism, providing services in this field and creating a material base for tourism. In its humanistic form, it focuses on the person himself as a participant in a tourist event. This anthropocentric accent does not contradict the idea of sustainable development of tourism and recreation, i.e., in relation to the human and natural environment [3–5].

What is the use of medieval monuments of the art of fortifications today? Well, it is the most common educational and tourist use. It belongs to the areas of cultural tourism and the phenomenon of martial arts tourism [6,7]. Let us try to look at these issues from many aspects.

The knight's castle and the knightly ethos are a remnant and, in a way, symbols of the traditions of medieval Europe, especially of the Western civilization. What remains to this

day? Could they have been preserved only in medieval poetry and drama [8]? Of course, numerous magnificent castles, the ones that have never been destroyed or rebuilt, are the pride of individual countries and a significant tourist attraction—A place of museums and various cultural events. However, in many cases, this Christian, chivalrous "spirit" seems to have disappeared. For members of knight brotherhoods operating at these facilities or for participants of knightly tournaments currently held, this is a hobby and entertainment, not service in the name of principles and values [9–11].

Martial arts, military knowledge and defence architecture arise from a specific cultural tradition, in a given context of technological possibilities and the environment of a specific material culture and dominant ideas. Therefore, in China, martial arts were developed especially in Buddhist and Taoist temples, and in Poland, mainly in castles and noble courts. Today, the destinations of martial arts tourism are often such places (Shaolin temple in China, other famous training centres, or castles, next to which enthusiasts of Western martial arts train in knightly fights [12].

In facilities used to cultivate a given cultural tradition, an important role is played by construction, the external appearance of the building's body, functionality, as well as interior design, including appropriate symbolism used in decoration. These are different cultural codes that make it possible to recognize the self-identification of users [13–15]. Thus, in some cases, defensive architecture can be equated with the architecture of martial arts. In each case, however, it is a cultural heritage that is of interest to cultural tourists [6,16,17].

Embankments and walls were built in distant antiquity. On the other hand, the fortified settlement was later turned into a stone castle with two or three lines of defence. The bergfried, or high castle, was usually the hardest to get. Indeed, for several centuries, the mighty stronghold was the security of its defenders. Only the popularization of artillery resulted in the decline of the strategic importance of defensive castles. The location of the castle in the area played a significant role—often on the top of a mountain or hill. This provided good visibility at greater distances and made it difficult for the enemy to access. Sometimes castles were built on the seashore, rivers or surrounded by a deep moat, with the use of a drawbridge.

Originally, castles were built by princes and kings. Then, after the formation of a feudal social hierarchy, counts and richer noble families also built defensive castles. The castle confirmed the position of a given family. With time, this knight's castle became part of the archetypal image of Europe. Driving through some countries by car, we can admire castles on many hills, blended into the landscape.

For several thousand years, Polish lands have been inhabited by militant cultures, which originally built earth and wooden embankments and defensive castles, sometimes also stone walls. These were Lusatian culture castles, such as Biskupin, or, for example, Scythian ones, such as the stronghold in Chotyniec [18]. The Ario-Slavic peoples, with the social structure called war democracy, used horses and carts very early and organized distant expeditions and aggressive military actions. However, the Lechite people, who created the Lusatian archaeological culture, were so attached to the occupied land that they developed the construction of defensive strongholds. The results of anthropological, linguistic and genetic research confirm it [19–21]. However, some archaeologists and historians disseminate the version of the late appearance of the Slavs in this area. For example, Koper [22] states that the Lechites did not reach today's Poland until the 7th century AD.

Undoubtedly, the stronghold in Biskupin, attributed to the Lusatian culture, "was surrounded by a vertical embankment of the so-called chest or chamfered structure: tall and strong boxes made of logs placed on the framework, filled with compacted earth inside. The same method of fortifications will be found in Slavic fortifications several hundred years later. The Eastern Slavs were particularly fond of this type of fortification [23]. Jasienica says after Ibrahim ibn Jakub (10th century) that: "The Slavs built most of their fortresses this way. They deliberately go to meadows abundant with water and thickets, and then draw a circular or quadrilateral line there, depending on the shape of the hillfort and the

area of its surface, they dig around [the ditch] and pile up the excavated soil, strengthening it with boards and a tree on the likeness of the ramparts, until the wall [rampart] is of the size they want. And they measure out the gate on which side they desire, and one enters it on a pier made of wood" [23]. This type of wooden pier was also built for the strongholds in Biskupin, Ostrów Lednicki, and Ostrów Tumski.

According to Koper, prince Mieszko I resided in Ostrów Lednicki, where one of the first stone castles in Poland was built [22]. At that time, the main wooden strongholds were built. Words: gard, grad, or hrad mean the same as a defensive stronghold, a castle or a city. In the pre-Piast period, the tribes of the Lechite Slavs living to the west of the Polans' seats probably had a dominant position, as evidenced by both the large number of castles and Czekanowski's research [19]. "Lech founded Gniezno, Szczyt-Szczecin, and another Lechite prince, Kiev-Kiev. Many Lechice castles and strongholds were built in what is now eastern Germany" [21,24].

From the middle of the 10th century, an interesting innovative change was introduced in the towns of the Polan country (reign of Mieszko I). Namely, a so-called hook structure was used. Well, the logs located transversely to the embankment run were equipped with huge catches that prevented the beams from sliding out of the upper, longitudinal layer. The structure thus became more stable. Jasienica [23] stated unequivocally that "the Old Slavic fortification system was completely separate and not at all similar to foreign patterns". Let's call it a "design innovation of Mieszko I".

The times of the Piast rule in Poland translate into material cultural heritage and its use in tourism, which is reflected in the tourist Piast Trail [25]. However, only the Polish king Casimir the Great (1310–1370), the last king of this dynasty, funded numerous stone castles. The so-called Eagle's Nests were guarding the main routes [26,27]. Stone castles were already built to the standards common to Western defence architecture of the time.

After the Polish–Lithuanian Union, there was a development of fortification construction in the areas of today's Belarus and Ukraine. An example is the family castles of King Jan III Sobieski (1629–1696) in Olesko and Żółkiew [28]. The castle legend, described in Henryk Sienkiewicz's Trilogy, namely in Kamieniec Podolski, is also now located in Ukraine [29].

Sustainability is a concept that has arisen from an increase in environmental awareness; this, in turn, translated into new, scientific areas of penetration and economic canons of conduct. This applies in particular to functioning in harmony with nature and caring for the natural world but also to anthropogenetic areas—created by man (parks, green areas). Buildings, such as historic fortresses and monasteries, built on hills and islands, integrated into the landscape and nature, are the object of ecological care for cultural and natural heritage [30], according to the canon of sustainable tourism and tourist use of similar places.

The main scientific problem here is to present the essence of the phenomenon of medieval fortified castles, especially Polish ones. This article is specifically about caring for cultural heritage in the sense of being sustainable in this regard. Thus, it is a discussion of the cultural phenomenon of medieval castles, with examples, and of the importance of this phenomenon for today's tourists.

Currently, there are over 400 castles or their ruins in Poland. The list of the most beautiful castles in Poland endorsed by Minister Gliński [31] presents 39 castles, including at least 25 (64%) defensive structures. The towns with castles are in alphabetical order: (1) Baranów Sandomierski, (2) Bobolice, (3) Brzeg, (4) Bytów, (5) Chęciny, (6) Głogówek, (7) Golub-Dobrzyń, (8) Gołuchów, (9) Goraj, (10) Grodziec, (11) Kliczków, (12) Kórnik, (13) Kraków, (14) Krasiczyn, (15) Lębork, (16) Łańcut, (17) Łęczyca, (18) Łubianka, (19) Malbork, (20) Nidzica, (21) Niedzica, (22) Ogrodzieniec, (23) Pieskowa Skała, (24) Rydzyna, (25) Sandomierz, (26) Słupsk, (27) Sucha, (28) Szczecin, (29) Sztum, (30) Szydłowiec, (31) Świdwin, (32) Tykocin, (33) Ujazd, Krzyżtopór, (34) Uniejów, (35) Wałbrzych, Książ, (36) Warszawa, (37) Wiśnicz, (38) Zagórze Śląskie, and (39) Zielina, Moszna.

Polish castles, including the Teutonic Knights, are not places of violence and oppression but victories in defence wars and the glory of the Kingdom of Poland, from the 15th century

in the union with the Grand Duchy of Lithuania. Poland did not wage partition wars. There are medieval gothic brick castles in Poland, including the largest one in Malbork. The castles are used for tourism as part of established tourist routes [32–36].

## 2. Literature Analysis and Case Studies

In the literature on the subject, we find the work about "three Polish castles: the Royal Castle in Warsaw, which was rebuilt completely after World War II, the Bobolice Castle, which was rebuilt after several hundred years of being ruined and the fake of the medieval Pszczyna Knights Stronghold, created recently" [37]. The issue concerns the authenticity of the castle as a tourist attraction. In our study, all castles are historical or rebuilt in the same way as the existing ones. This authenticity is a value appreciated by cultural tourists aware of the national history [38–40].

Additionally, tourists from outside Europe, such as the Japanese, look for authentic, historical places associated with chivalry and important events or images from mass culture (manga, cinema) [41–43]. Europe's fortified castles fascinate tourists from many countries around the world. Objects related to the military and religious tradition of Asia, such as the Shaolin monastery, similarly attract tourists [44,45].

The analysis of tourists' motivation most often indicates the willingness to meet monuments of historical, national, and cultural heritage [36,40,46], which is used for the optimal management of castles as museums [26,27,36].

What does the presence of Polish castles look like among the works of defence architecture in Europe? In the album Palaces and Castles in Europe by the German publishing house Du Mont Monte we find only the castle in Malbork, which is presented as German [47]. Meanwhile, it was indeed built by the Teutonic (not German) state, but it was bought by the Polish king and has been a Polish castle for centuries. In particular, it was at one time the largest European brick castle and was the most modern building in terms of the medieval art of fortification [27]. The castle in Malbork is one of the buildings in today's Poland, also included in the work Castles of the World [48].

The castle in Malbork, Poland, is perhaps really the most interesting. The Battle of Grunwald and the siege of Malbork Castle-reconstructions and stagings take place each year in July. In addition, archery competitions, exhibitions of siege machines, and exhibitions of medieval armour and weapons, etc., are organized in the castle [27,32,33]. The Castle Museum in Malbork is a significant museum and educational facility in Europe. The castle, which was originally the capital seat of the Teutonic Order, has been relatively well rebuilt and attracts crowds of tourists from many countries around the world.

Kamieniec Podolski, mentioned above, is a kind of symbol of the power of the first Polish Republic. A castle high on a solid rock seemed impregnable. He guarded the eastern frontiers, known as the borderlands of the Polish kingdom. In the monograph of martial arts tourism, in the section devoted to the old Polish tradition of martial arts, two pages and three photos are devoted to this [12].

There is a castle in Khotyn (in Polish: Chocim) on the Dniester nearby; built in the 14th century, it guarded the south–eastern border. It was the site of three victorious battles for Poland (1509, 1621, 1673). Today, the castle is a museum object in Ukraine, just like Kamieniec Podolski and the castle in Olesko [28].

In turn, for example, Kamieniec in Odrzykoń (Podkarpackie Province) is a ruin that will probably not be rebuilt [49]. This 16th-century defensive castle belonged to King Casimir the Great and later to Mikołaj Kaminiecki, the Great Crown Hetman (1502), among others.

A curiosity is the "Castle" in Piotrków Trybunalski (Schemes 1 and 2), which has a proud name—the Royal Castle. It was built in the years 1512–1519 by master Benedict from Sandomierz on the order of the Polish king Sigismund I the Old. It is a residential tower in the Gothic Renaissance style, four-storey, and built of brick and sandstone. Currently, it is the seat of the museum. Although some inhabitants of Piotrków are not sure that there is a castle in their city, some of them are able to properly guide a tourist interested in

seeing this building. In this historic city, the Crown Tribunal met in the 16th century. On the other hand, the tower in the centre of the city, despite its thick walls, did not have much defensive but prestigious importance. The specific Lechic stronghold described above, with the design innovation of Mieszko I, was an advantage of the first Piasts. Forts built by the ancestors of today's Poles in times earlier by several or a dozen or so centuries also deserve more attention. Another contribution of Poles to the art of fortification was the so-called bay windows, which were used in numerous castles built by Casimir the Great. The same king also built many brick cities.

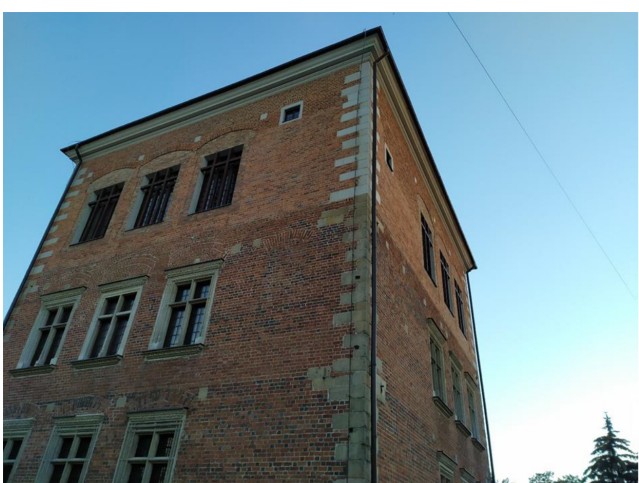

**Scheme 1.** Castle in Piotrków Trybunalski—main elevation [made by one of the authors].

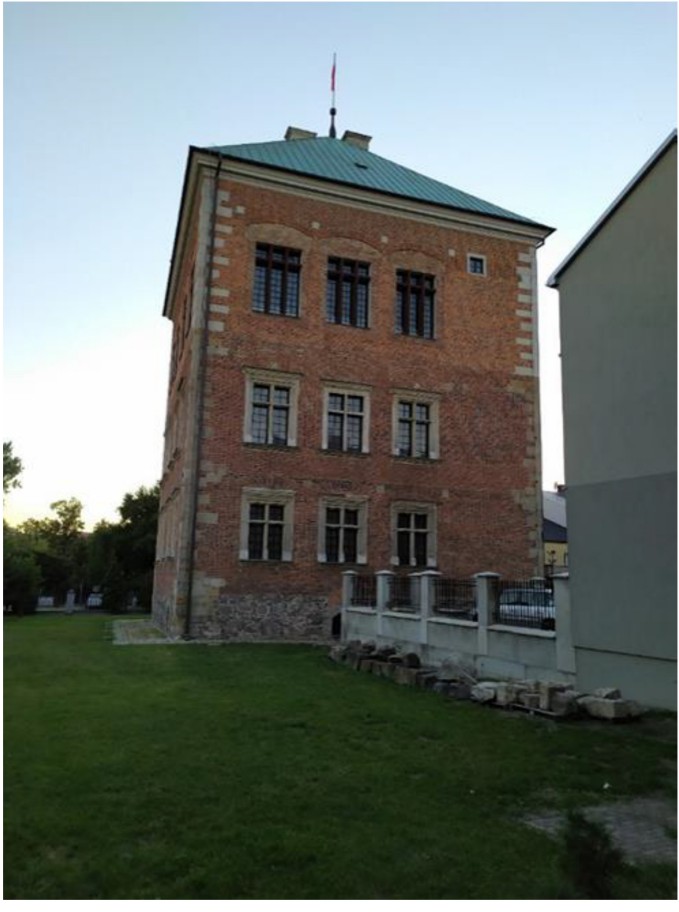

**Scheme 2.** Castle in Piotrków Trybunalski—another shot [made by one of the authors].

After referring to five Polish castles, let us recall other buildings of this type. The selected castles of Europe included here are a deliberate selection. These are the fortified buildings that delighted or amazed the author of this study idea. With considerable difficulty, the author proposed a series of 10 strongholds (including one famous monastery) (Scheme 3). In this way, we find a relatively subjective choice—A study of 10 cases of objects that may be differently attractive to Poles, other Europeans, and people from outside Europe.

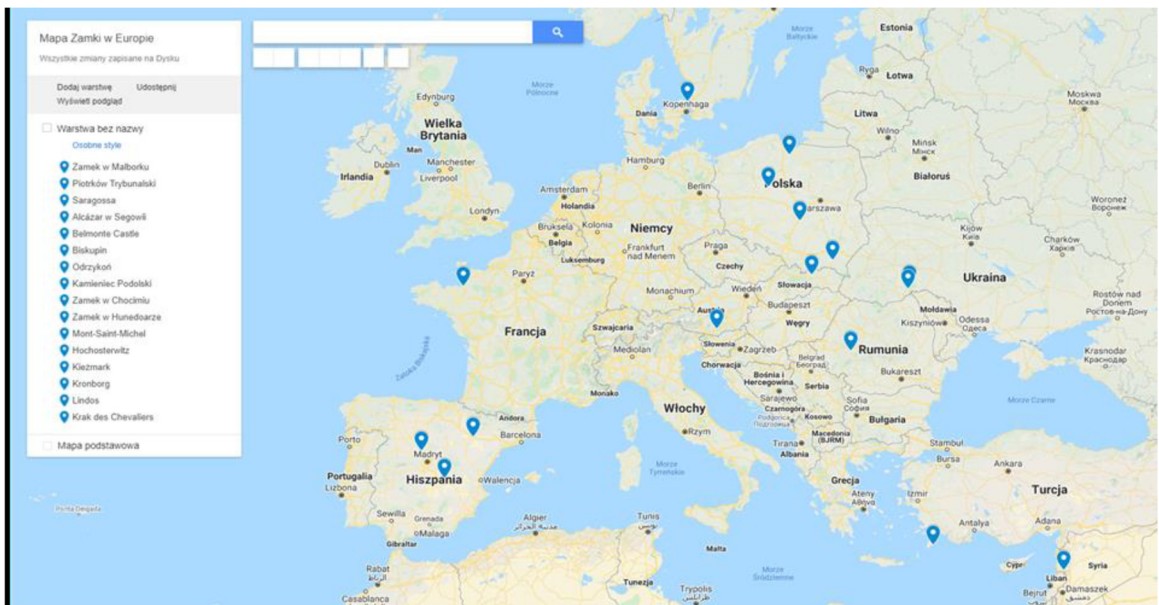

**Scheme 3.** Listed castles in Europe and the Middle East [own study based on Google maps].

1. Alcázar de Segovia, Spain. This building (Scheme 4) was built to be defensive, but in its history, it served as a royal palace, a prison, and the seat of an artillery school. The first records of its existence date back to 1120, but it is very likely that the first buildings appeared here in antiquity. During archaeological research, granite blocks similar to those used to build Roman aqueducts were found. The fortress was rebuilt many times by successive owners and adapted to their needs. In the castle, you can admire, among other things, a collection of weapons. One of the courtyards enters the richly furnished royal chambers. Available to visitors is a throne room, fireplace room, royal lounge, bedroom, and chapel, among others. The facility is open year-round.

2. Castillo de Belmonte, Spain is an impressive gothic castle built between 1456–1468. There is also a museum in the castle. El Cid was recorded here. It was the site of the first IMCF World Championship, held 1–4 May 2014. IMCF (International Medieval Combat Federation) was founded in 2013. The Championship—knights' struggles from many countries around the world—was watched by over 10,000 spectators. It is a completely new sport, however, referring to the medieval European tradition of knight tournaments. Fights are played, among others in 1 vs. 1, 5 vs. 5, 16 vs. 16. In this tournament, knights from the USA won in the overall ranking, followed by Poles.

3. Corvin Castle (Hunyad Castle, Hunedoara Castle) in Hunedoara, Transylvania, Romania. It was built by the Hungarian nobleman Janos Hunyadi in the 15th century. It is a unique monument of military culture—the art of fortifications. Its architecture is Gothic Renaissance. Due to its historical heritage, it is most interesting for Hungarians and Romanians but is also the pride of the famous (not only thanks to the film count Dracula) of Transylvania.

4. Hochosterwitz (Ger. Burg Hochosterwitz, in Slovenian Grad Ostrovica), Karyntia, Austria. It is a medieval fortress (Scheme 5) that has never been conquered. It is situated on a 175 metre-high dolomite rock. A 620 metre-long road leads to the castle through

14 defensive gates, added in the 16th century. An alternative road also leads to the summit, the so-called path of fools. It is so steep that it is closed for the safety of tourists. Visibility from the tower is approximately 30 km in good weather. There is a special cable car for tourists. The castle houses a museum dedicated to its history and a gallery of portraits of the members of the Khevenhuller family, who have owned the fortress for four centuries, the armoury and the chapel of St. Nicholas.

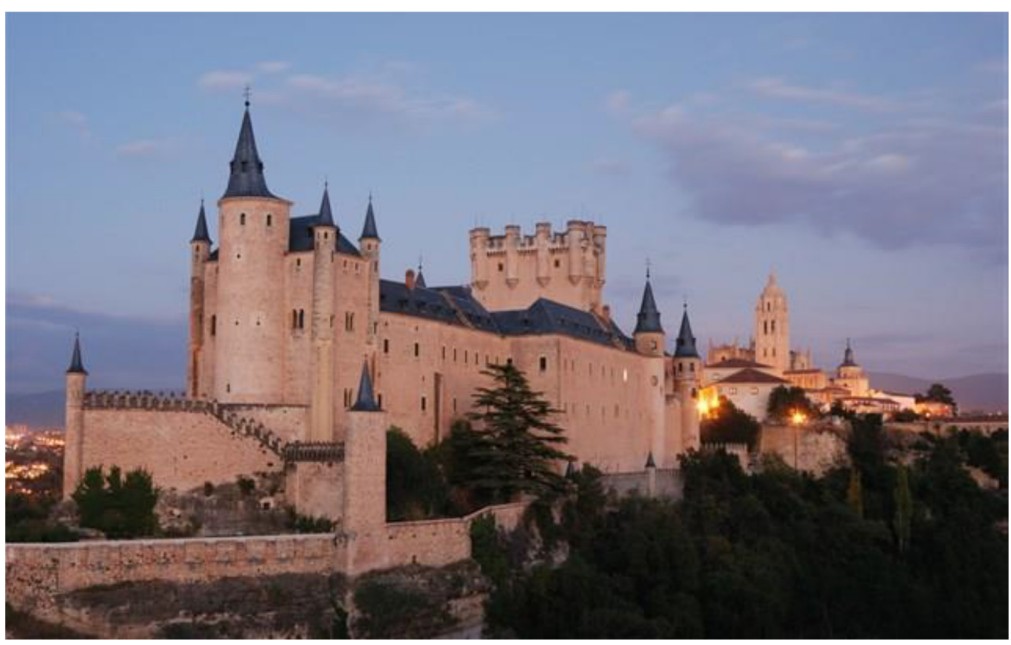

**Scheme 4.** Alcázar Castle in Spain [50]. View from the South side.

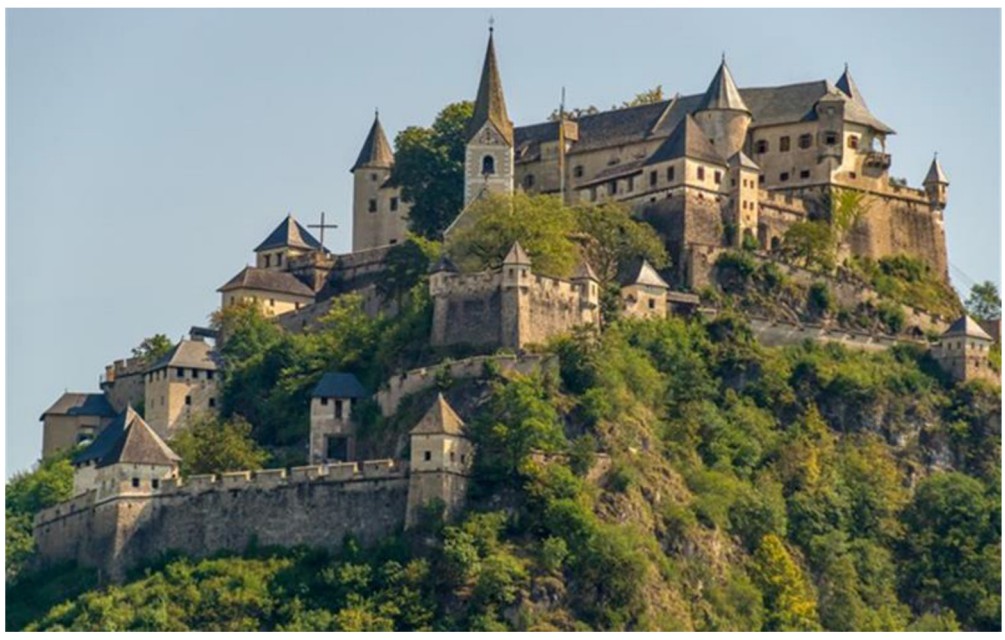

**Scheme 5.** Hochosterwitz Castle, Austria [50].

5. Kežmarok (Spisz, Slovakia) city castle (connected with the city walls), gothic from 1460–1470. Built by Hungarians from the Zapolya family, it was alternately in the hands of Polish and Hungarian noble families. In 1702, the city bought it. It is now home to a museum. It is an interesting case of city fortifications in terms of tourism.

6. Krak des Chevaliers (or Crac des Chevaliers, "Fortress of knights")—a castle in the western part of Syria, in the Jabal al-Nusajrijja mountains, the former seat of the Knights Hospitaller. The castle was built in the 11th century and expanded in the 12th century. The outer walls of the lower castle are 6 metres-thick. Stocks sufficient for five years were constantly kept in the fortress. It was a model fortress in its time and has been on UNESCO's World Heritage List since 2006. It is today one of Syria's top tourist attractions.

7. Kronborg Castle, Helsingor, near Copenhagen, Denmark. It is situated on the Øresund Strait that separates Denmark from Sweden. It has been operating since the 15th century. The Renaissance Castle on UNESCO's World Heritage List is also promoted as "Home of Hamlet" and the castle of Danish kings.

8. Lindos, on the island of Rhodes, the castle of the Knights Hospitaller. The Knights of the Order of St. John rebuilt and enclosed the ancient Acropolis with a Temple of Athena. The high walls rising from the sea level are delightful.

9. Mont Saint Michel—St. Michael the Archangel, a fortified male Benedictine monastery from the 8th century on a peninsula that becomes an island at low tide. A stronghold on the border of Normandy and Brittany (France), it has been on the UNESCO World Heritage List since 1979.

10. Zaragoza Castle—The Aljafería Palace in Zaragoza, Spain. It is another monumental building from the time of the battles for Christian Spain. It was built in the 11th century as an Arab castle and palace by Al-Muqtadir. In 1118, Zaragoza was finally conquered by the Christians, more precisely by the army of the king of Aragon and Pamplona, Alfonso I the Brave (Alfonso I el Batallador). From then on, this beautiful castle became the new seat of Christian kings.

We omit here great, beautiful castles and palaces on the Loire, such as the Castle in Chaumont (Fr. Château de Chaumont)-in Chaumont-sur-Loire in France, built in the 10th and rebuilt in the 15th century, and the Malbork Castle—the largest brick castle in Europe.

What did Poles contribute to defensive architecture and the art of fortifications when it comes to building castles? Casimir the Great was the initiator of the fact that "bay windows were used in several dozen Polish castles—A cheaper and equally effective form of wing defense of walls. This solution, also used here in city fortifications, has become a distinguishing feature of the Polish art of fortification" [15,27].

## 3. Methodology

The scientific problem undertaken is the importance of castles for the functioning of cultural tourism in the opinion of the inhabitants of Central Europe, which is also the topic of the questionnaire used in the research. The main method of research is a diagnostic survey carried out with the use of a survey on a group of *n* important according to the statistics in the number of *n* = 614 respondents. A template of the questionnaire is in the Appendix A attached.

The selection of respondents included students, mainly from the faculty of "tourism and recreation", and other tourists, including members of reconstruction groups. They were especially people potentially interested in history and cultural heritage.

In order to answer the research questions and test the hypotheses, statistical analyses were performed using Statistica version 13.3. Basic descriptive statistics were calculated on the basis of which histograms were created in order to present the research results in a deeper and more precise way.

The research concerns women and men in the age range according to statistics from 18 to 74 years of age with primary, secondary, secondary, and higher education. The survey was attended by respondents from Poland, Ukraine, and several other European countries

In the process of selecting the sample, the activities of the statistical population with people studying in Poland and Central Europe were used. The data presented in this article was obtained on the basis of the following charts and sample research questions:

1. Was the trip to the castle/castles the main purpose of your trip?
2. If a stay in a castle is one of the elements of a tourist trip, how can you describe it?

3. Please indicate the main motive for your trips to the castle(s).
4. What additional attractions, do you think, should accompany visiting the castle?
5. Which castles from the mentioned periods have the greatest tourist value for you?
6. Which of the visited castles in Europe, according to you, is the most interesting in terms of tourism?
7. Were these objects visited by the way, or was the trip undertaken specifically to see the castle?
8. In your case, was it related to your interest in the history and culture of old eras, or perhaps with the traditions of military and martial arts?

Observing the results of the research presented in the histogram Figure 1, we can notice a very large discrepancy in terms of the age of the respondents, which is somewhat diversified, and also increases the truthfulness of the research results.

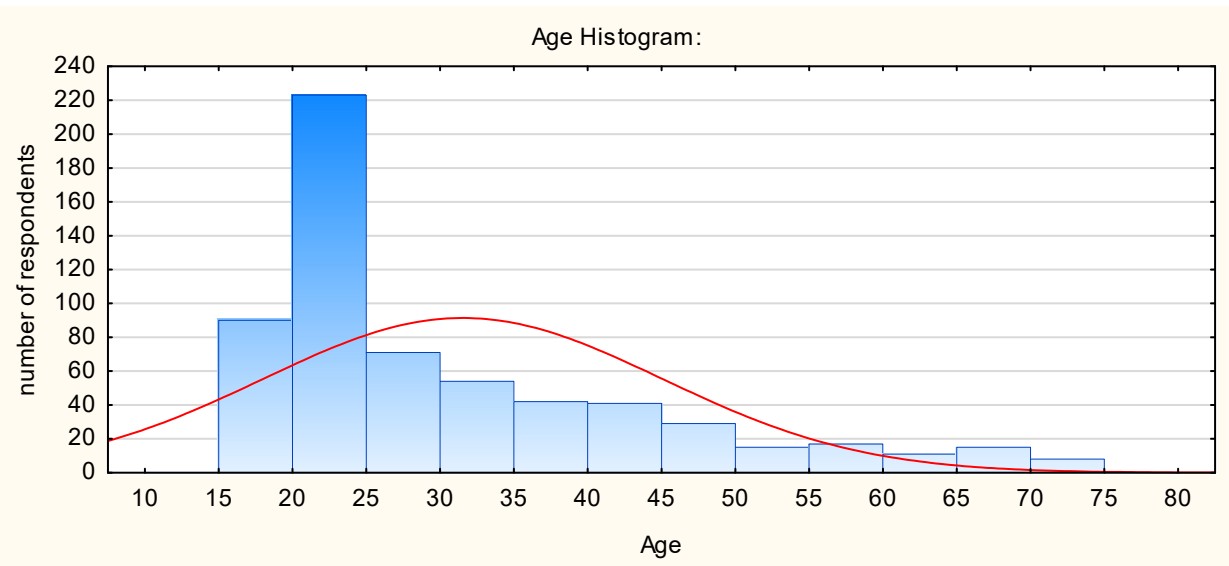

**Figure 1.** The chart shows the categorization of the variable age.

The age range of the respondents is between 18 and 74 years old. The observations are evenly distributed over the intervals which means that the ages vary. The dominant age range of the respondents is between the ages of 20 and 25.

In the presented histogram Figure 2, we notice that the majority of respondents concern women, which is equal to 394 cases, while men are present in 220 cases.

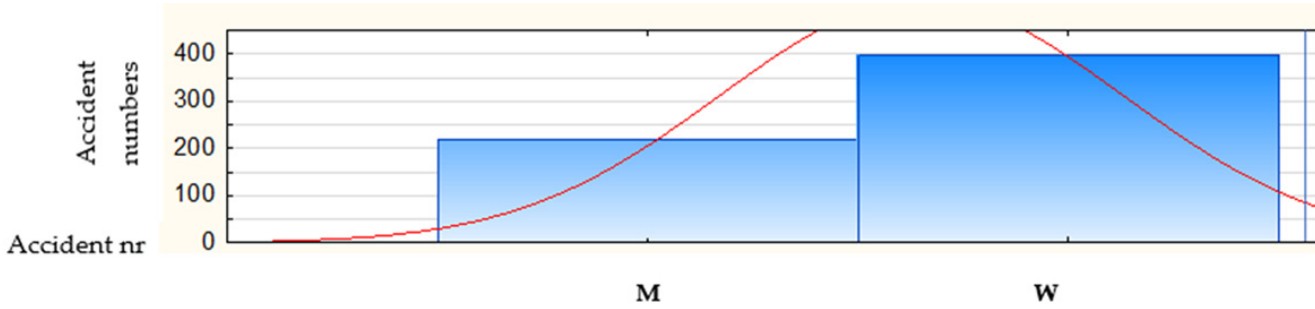

**Figure 2.** The graph shows the categorization of the variable gender.

The histogram shows that the majority of respondents are women aged 20–25. The situation is similar among men, with the difference that there are fewer male respondents, which is shown more tenderly in the Figure 3.

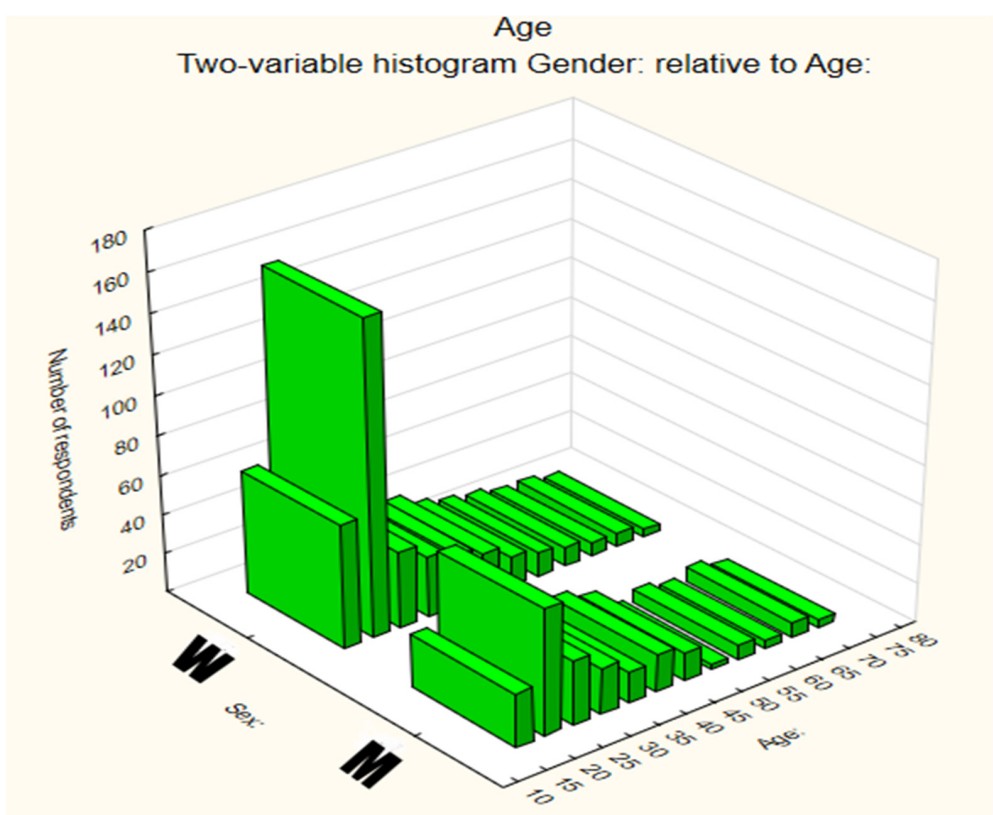

**Figure 3.** The graph shows the statistical relationship according to the variables of age and gender.

On the histogram Figure 4, we observe the number of respondents in terms of education. The results of the research indicate the following: primary education—12 cases, secondary education—313 cases, and higher education—289 cases. From the results of the research, we observe that the largest number of respondents, 308, represents the group with secondary education.

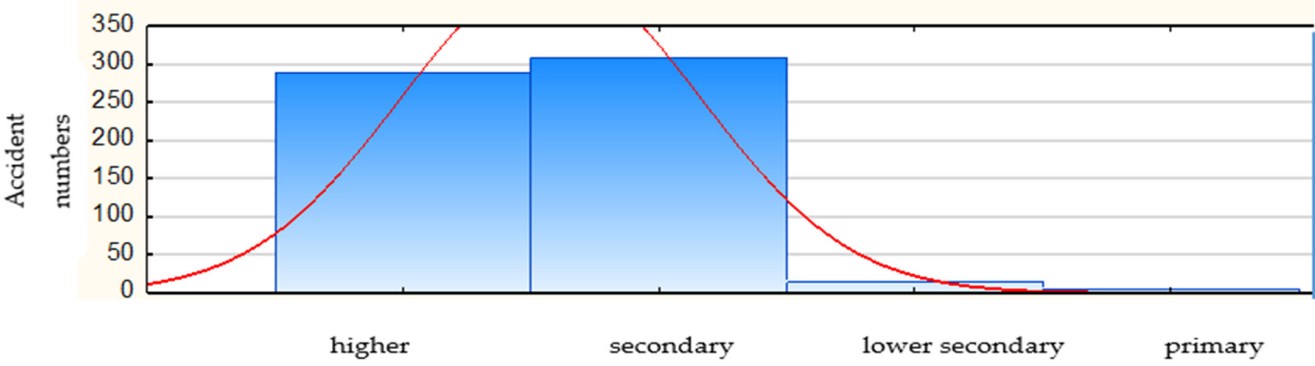

**Figure 4.** Graph for the typological distributive series for the variable qualitative level of education.

The results of the research in a histogram Figure 5, present social groups in terms of their occupational status. The observations were given to the respondents in an even manner, which in this case means a wide professional range. The chart shows that the occupational positions are as follows: working—262 cases, studying—271 cases, disability pensioner—7 cases, pensioner—28 cases, and unemployed—46 cases.

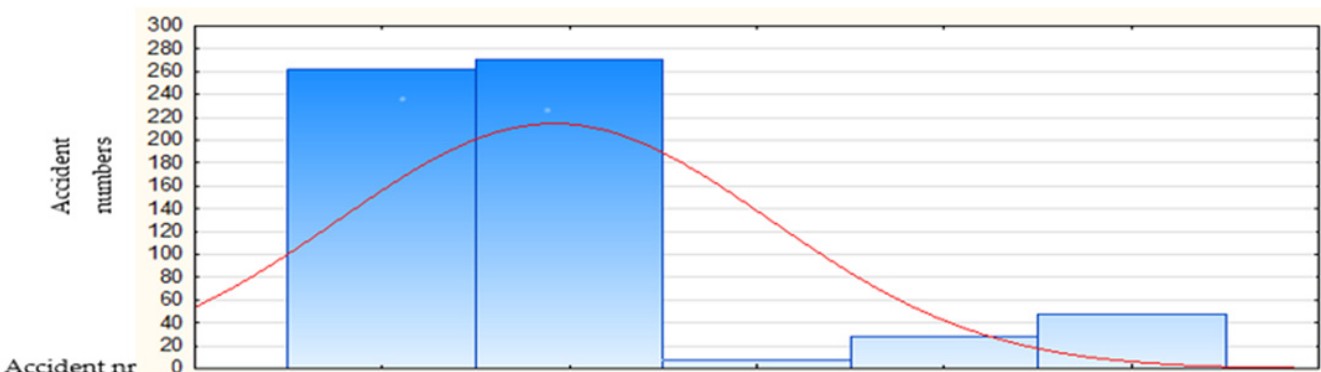

**Figure 5.** Graph for a distributive series, typological for the variable relating to the occupational position.

## 4. Results

Initial data cleaning was performed at the stage of entering the answers into the spreadsheet. It was very important to create the same sheet for all introductory persons. By analysing the data presented in Table 1, it can be stated to what extent the results allow to notice irregularities in the data and to what extent the apparently correct values are burdened with errors resulting from unreliability or mistakes of the respondents or the interviewers themselves. The study involved slightly more women than men of different ages with an advantage between the ages of 20–25 and that most often were working and studying in secondary and higher education. Most of the respondents come from large cities with an international reach.

Moving on to the interpretation of the dependencies of the variables, we can say that the correlation of the age variable shows the significant and strongest relationship in the responses to the questionnaire questions: 1. Was the trip to the castle/castles the main purpose of your trip? if so, was it a trip to: country, region, or town, and where (?) 2. How many times was a trip to a castle or a group of castles the main purpose of the trip (please provide the number)? 3. Have you participated in tourist trips whose program included visiting castles? If so, was it a trip to a country, region, or town (?) 4. In your opinion, the castles of which historical epoch have the greatest tourist value? Medieval (Romanesque, Gothic), Renaissance, Baroque, Enlightenment, or 19th century. 5. Which of the visited castles in Europe are, according to you, the most interesting in terms of tourism? I am asking for comments about your trips, during which you saw medieval castles. It is a longer statement that will contain answers to the following questions: 5.1. Where did they go (countries, regions, or places) and how many times? 5.2. I am asking for a few general comments about such trips, their motives, goals, and impressions. Were these objects visited on the way, or was the trip undertaken specifically to see the castle? In the correlation of the variable concerning gender, a significant and strongest correlation can be observed in the answers to the question: "If your trips resulted from your interest in the traditions of military and martial arts, then they concerned historical reconstructions, museums and weapons shows, museums and shows of costumes from old eras, military architecture, Entertainment, curiosity". Observing the correlation between the variable representing the respondents' education and the answers from the questionnaire, it is presented in the following questions: 1. How many times was a trip to a castle or a group of castles was the main purpose of the trip (please provide the number)? 2. In your opinion, the castles of which historical epoch have the greatest tourist value? Medieval (Romanesque, Gothic), Renaissance, Baroque, Enlightenment, or 19th century. 3. Where did they go (countries, regions, or places) and how many times? Taking into account the correlation between the variable concerning the country of residence and the answers from the questionnaire, we observe a significant and strongest correlation in the responses to the question: "How many times was a trip to a castle or a group of castles the main purpose of the trip (please provide the number)"? When reading the relationship between the occupational position and the

answers to the questionnaire questions, the significant and strongest relationship is read in the responses to the question: "Were these objects visited by the way, or was the trip undertaken specifically to see a given castle? I am also asking for a few general comments about such trips, their motives and goals, impressions".

**Table 1.** Correlation between the variables concerning age, sex, education, country of residence and occupational position and the answers to the questions in the main part of the questionnaire.

| VARIABLE | Correlations The Determined Correlation Coefficients Are Significant with $p < 0.05000$ $n = 349$ (Missing Data Was Removed by Accident) | | | | | | |
|---|---|---|---|---|---|---|---|
| | Mean | Standard Deviation | Age: | Sex: | Education: | Home Country: | Professional Position: |
| Was the trip to the castle/castles the main purpose of your trip? | 3.50 | 0.5007 | −0.042295 | −0.022726 | 0.038828 | 0.002352 | 0.031880 |
| If so, was it a trip to: country, city, … | 270.15 | 132.9141 | **0.123682** | 0.059016 | 0.020619 | 0.003962 | 0.000815 |
| How many times have a castle or group of locks been the main purpose of your trip (please provide the number)? | 2.86 | 5.3855 | **0.234864** | −0.001295 | **−0.136086** | **0.153551** | −0.086062 |
| Have you participated in tourist trips where visiting castles was one of the items on the agenda? | 3.20 | 0.4010 | −0.026567 | −0.045235 | 0.051094 | 0.053372 | 0.033289 |
| If so, was it a trip to: country (s), city, … | 343.46 | 146.7596 | **0.140808** | −0.099301 | −0.078226 | 0.047746 | 0.031650 |
| If a stay in a castle is one of the elements of a tourist trip, how can you describe it? | 8.16 | 1.2181 | 0.016946 | 0.072687 | 0.072852 | 0.062363 | −0.004865 |
| Please indicate the main motive for your trips to the castle (s). | 34.97 | 3.6316 | 0.018676 | −0.034272 | −0.053104 | 0.036270 | −0.018878 |
| What additional attractions, according to you, should accompany you while visiting the castle? | 33.19 | 2.8525 | 0.047840 | 0.045042 | −0.012062 | 0.065158 | −0.032765 |
| What emotions did you experience while visiting the castle? | 10.47 | 1.2490 | **−0.140426** | 0.036471 | **0.116957** | 0.016877 | −0.012389 |
| Castles from which historical era do you think castles have the greatest tourist value? | 8.19 | 1.1307 | 0.052204 | 0.046361 | −0.064819 | 0.003834 | 0.009203 |
| If your trips resulted from your interests in military and martial arts traditions, they concerned: | 8.31 | 1.5133 | −0.025491 | **0.164703** | 0.046083 | 0.021147 | 0.010186 |
| Which of the visited castles in Europe are, in your opinion, the most interesting in terms of tourism? | 167.35 | 137.8653 | **0.141092** | −0.058563 | −0.010184 | 0.028385 | −0.008188 |
| Where did they go (countries or places), how many times? | 521.06 | 263.9548 | **0.111993** | −0.024936 | **−0.108646** | 0.000547 | 0.022536 |
| Were these objects visited by the way, or was the trip undertaken specifically to see a given castle? | 317.56 | 98.2604 | **0.115850** | −0.050356 | 0.025793 | 0.049143 | **0.114738** |
| In your case, was it related to your interest in the history and culture of old eras, or perhaps with the traditions of military and martial arts? | 504.84 | 166.7705 | 0.069986 | −0.018923 | 0.028253 | 0.051651 | −0.002414 |
| Which historical object (among the castles visited or viewed from outside) made the greatest impression on you and why? | 667.98 | 327.8082 | −0.011623 | −0.060403 | 0.053903 | 0.052740 | 0.074532 |

### 4.1. Data Validation

The results presented above did not raise any suspicions in the authors. During the preparation of the spreadsheet and performing the initial data processing, some problems

were omitted. Can you expect others that the authors were unable to capture? It turns out that the appropriate structure of the survey and the knowledge of the most common errors sometimes allows us to track the opponent, who in this case will be a dishonest respondent or an unreliable interviewer.

*4.2. Tourist and Educational Use*

Among cultural tourists visiting fortified castles, supporters of historical and military tourism prevail [17] as well as reconstructors and "historical experts" [6,17]. In particular, these are objects of interest to enthusiasts of knightly and medieval traditions [9–11].

Out of these 39 Polish castles mentioned above, seven (18%) are home to knight brotherhoods, shows, tournaments, or similar knightly events. In one case (Golub-Dobrzyń) there is also a "knight's school," which in particular conducts horse riding lessons [31]. In turn, the promotional brochure of the Polish Tourist Organization mentions and describes only 24 fortified castles. Indeed, some "castles" are palaces, not fortified buildings [51]. Only the castle in Golub-Dobrzyń was distinguished here as the one where knights cross swords [51]. On the other hand, in another promotional list, we find 13 castles, including fortified ones, such as the castle in Malbork and eight others, two royal ones (Warsaw and Wawel), a residence in Krasiczyn, and a palace in Łańcut, which once had bastion fortifications [52]. According to the town, we can find them here: Malbork, Warsaw, Krzyżtopór-Ujazd, Łańcut, Krasiczyn, Dunajec-Niedzica, Kraków, Pieskowa Skała, Ogrodzieniec, Będzin, Gołuchów, Książ, and Czocha. It is a kind of trail in the order given above, stretching from Malbork in the north of Poland, through Warsaw to the south and southeast to Krasiczyn on the San, and from there west to Czocha at the Czech and German borders.

Polish castles, although mostly not as impressive as Krak des Chevaliers, have been relatively well developed for educational purposes (museums and live history lessons, with the help of reconstruction groups) and purely tourist purposes (thematic routes) [25,26,53,54]. The same applies to the national heritage of other European countries for many years.

Particularly noteworthy is the ancient and early medieval history of Polish lands. The archaeological open-air museum "Carpathian Troy" in Trzcinica (from 2100 BC) and the stronghold in Biskupin (archaeological museum) [15,21] are tourist assets that are still underused.

What is the result of research/solving a scientific problem?

*4.3. Discussion*

So far, Polish castles have not enjoyed too much fame as attractions of international tourist traffic. Among 30 European castles and fortifications, there is not a single Polish castle in the work of the American researcher Terri Hardin [55]. Even more so, due to the inconsistency in the environment of Polish archaeologists and historians themselves, it is overlooked or underestimated by the Lechite, Western Slavs. Since around 2000 BC, their castles have been recognized unequivocally as Slavic by the German researcher Harald Haarmann [55,56]. However, many Polish scientists perceive them as Celtic (?) or Germanic (?).

Cultural exchange in antiquity and the Middle Ages was often done using brutal methods, fire, and sword. This is how Alexander of Macedon, the Romans and the steppe nomadic peoples of Asia transferred their culture and technique. The winner usually imposed his culture. To secure possession, he built walls and other fortifications. The elements of this kind of cultural dialogue were fortified castles and other buildings of defensive architecture. In Poland, we visit today mainly Polish castles but also Teutonic castles—the remnant of the defeated monastic state. In turn, Polish castles are located in today's Ukraine, being a remnant of the splendour of the Polish–Lithuanian Commonwealth–Poland in the union with Lithuania. The goods of the Lithuanian nobility include today's Belarus. In turn, Hungarian castles are located in Slovakia and German castles in the Czech Republic, etc. Together, they create the historical and cultural wealth of Central Europe. The same applies to other areas of Latin West culture, such as the castles of the Knights Hospitaller.

Some ruins of Polish castles in today's Poland and Ukraine are worth rebuilding. The situation is better in Lower Silesia and the Kłodzko Valley. Perhaps in many cases, however, it would be appropriate to invest, restore, and sell or donate to passionate individuals or associations. These facilities could serve even better cultural tourism and historical education, for example by running "knight schools" (ethos, equestrianism, fencing, other martial arts). It is an additional attraction for tourists [11,16,57].

The aim of the research was the importance of castles for the functioning of cultural tourism in the opinion of the inhabitants of Central Europe. The study was conducted in a group of men and women. The analysis of the obtained data showed that women are more interested in tourism. When comparing the average of tourists for a different type of travel, between women and men, it can be concluded that the situation is exactly the opposite. This is a very interesting phenomenon that should be further investigated towards the divisions of tourism. In the future, additional variables should be taken into account in the direction of the synthesis of temporal and spatial phenomena related to tourism and tourism geography. Perhaps this relationship is accidental and longitudinal studies should be carried out to investigate it. It is also possible to correlate the subjective quality of life in a given environment, bearing in mind different interests and responsibilities. Another very important issue that is often overlooked in research is how to measure the quality of life. Due to the multitude of questionnaires, the research conducted so far is very different in terms of methodology.

The following results were derived from the conducted research:

Based on the histogram Figure 6, it can be noticed that by asking the respondents the question: "Which of the visited castles in Europe, according to you, is the most interesting in terms of tourism?" Bearing in mind that the choice was very diverse, the results of the research are as follows: 226 respondents mentioned Pena Palace, which is the dominant feature in this case, in 61 cases the selection concerned Książ Castle, Schönbrunn Palace–68 cases, Chatelet–24, Alcázar–29, Hohenzollern Castle–25, Eilean Donan–112, and Ujazdowski Castle–69.

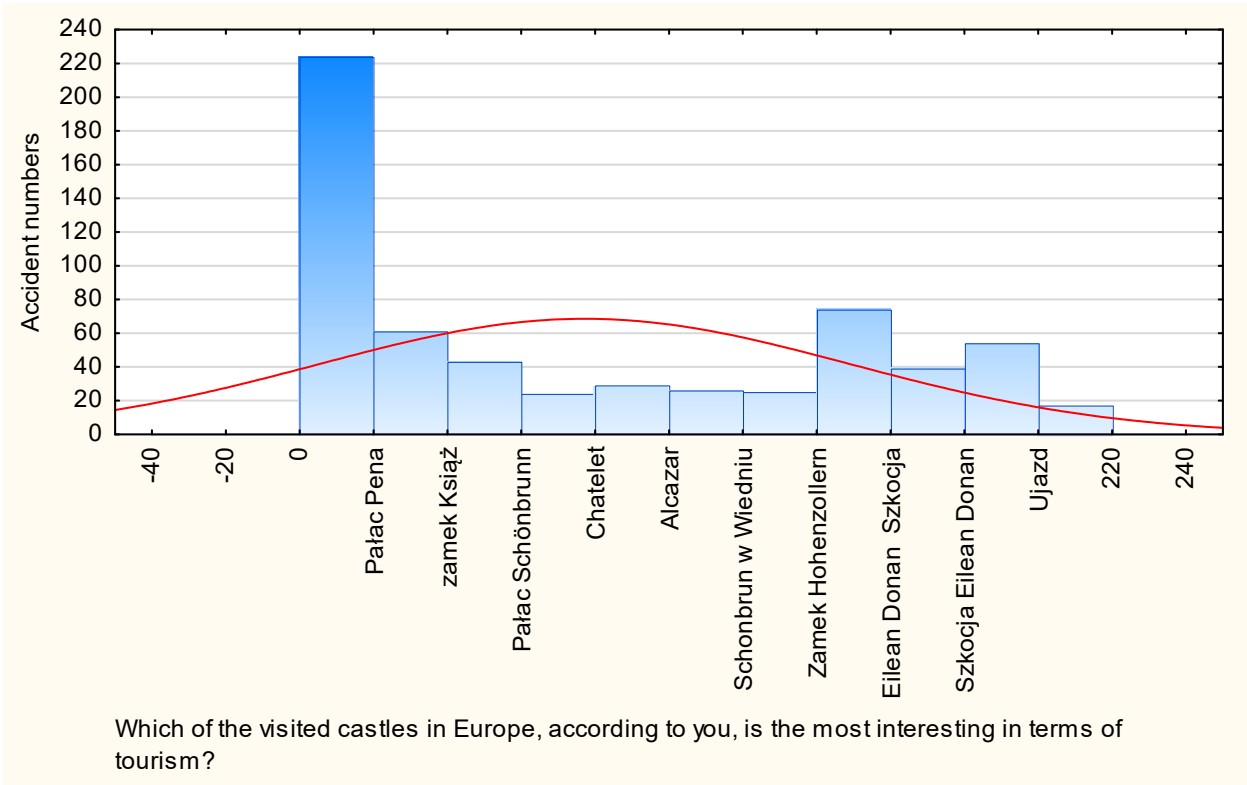

Which of the visited castles in Europe, according to you, is the most interesting in terms of tourism?

**Figure 6.** The most interesting castles.

In the case of the question: "If your trips resulted from your interest in military and martial arts traditions, they concerned (what)?".

The results presented in the above histogram Figure 7, show that the most numerous range among the respondents in the number of 197 cases, which means the dominant, was presented in the answers concerning entertainment and curiosities. From further research results, it can be observed that the responses in 136 cases concerned museums and shows of costumes from old eras, historical reconstruction in 109 cases, military architecture-102 cases, and 70 cases related to museums and weapons shows.

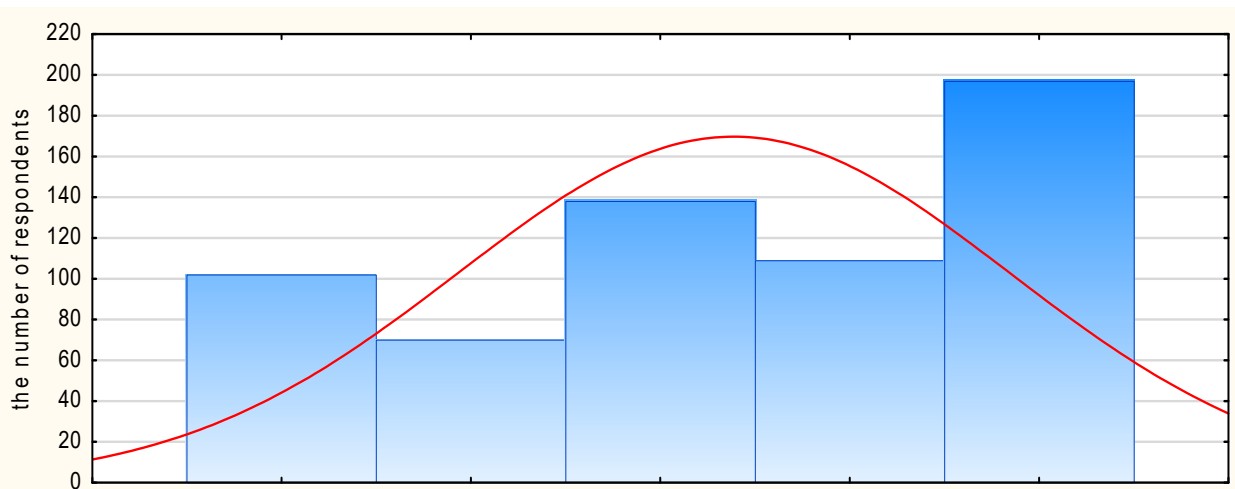

**Figure 7.** Tourist interests. Military museums and historical entertainment, interesting facts and weapons old costumes reconstructions.

This is similar to the sustainable use of heritage as it is realized and researched in the case of China, its martial arts traditions, and related objects [58–60].

## 5. Summary

The innovative approach of the authors is based on the study program used in the development of "Statistica" version 13.3 for data from various sources on a European scale. This made it possible to precisely define the goals of the trip, regarding sightseeing and supplementing the knowledge, as well as meeting the needs of learning about history, as well as getting acquainted with the prevailing historical tradition in castles in Europe. The questions presented in the questionnaire have a universal dimension, with particular emphasis on the interests of the importance of castles for the functioning of cultural tourism in the opinion of the inhabitants of Poland/Central Europe. In addition, the interests and purpose of the trip as well as the purpose of visiting castles by the respondents also concerned the desire to deepen historical knowledge, interest in historical architecture, reconstruction of historical battles with the use of various martial arts, as well as interests related to culture and art, craftsmanship and more down-to-earth activities, such as tasting dishes and drinks. Other researchers of this issue indicate similar attitudes and motivations [17,36,53,57,58]. Here, however, a greater accent fell on the specificity of castles preserved in Poland. Many respondents point to their historical interests as additional "chivalric attractions" [57].

For future research on personality correlates of tourism, it is recommended to conduct research on a larger sample and make more advanced calculations. It would be worth dividing the respondents into three groups depending on the level of physical activity and lifestyle: sedentary, recreational, and athletic. Accurate determination of the relationship between the demand and the pleasure of sightseeing could contribute to the development of effective methods to motivate tourism. In this respect, it would be worth assessing to what

extent the gender differences in the personal correlates of sightseeing have psycho-physical and social determinants or whether they are the result of the interaction of given indicators.

Further research should include representatives of a larger number of European countries and comparatively selected non-European countries. Since defensive castles, as a specifically European cultural phenomenon, deserve special anthropological and cultural research, they should be carried out in interdisciplinary teams.

## 6. Conclusions

There is a noticeable increase in awareness of broadly understood tourism. The respondents are able to precisely express their expectations in this area. Undoubtedly, also on the part of the tourist transport organizer, such as a tourist office, the need to expand the sightseeing area and make interesting objects available not only in Europe but also in the world is of key importance.

In a few cases, the age of the respondents determined the answers to the questions about trips to visit castles and in one case, their gender—when the question concerned the relationship with military interests and martial arts. Men are more likely to show this type of interest.

Another conclusion resulting from the survey is the high degree of trust that the respondents show in relation to the choice of the object which, in their opinion, is the most interesting in terms of tourism. These opinions raise the rank and prestige not only of historical sites but also of broadly understood tourism, as well as respect for the work performed as a pilot and tourist guide.

The great interest in the willingness to fill in the questionnaire by the respondents in comparison with the obtained research results indicates that the issues of tradition related to the military and martial arts are very important for visitors.

**Author Contributions:** Conceptualization, W.J.C.; methodology, J.K.; software, J.K.; validation, W.J.C., J.K. and P.Ś.; formal analysis, J.K. and W.J.C.; investigation, W.J.C.; resources, W.J.C. and P.Ś.; data treatment, B.P., A.R. and P.R.; writing—preparing an original project, W.J.C. and J.K.; writing—review and editing, W.J.C.; visualization, W.J.C.; supervision, W.J.C. All authors have read and agreed to the published version of the manuscript.

**Funding:** This study did not receive any external funding.

**Institutional Review Board Statement:** Not applicable.

**Informed Consent Statement:** Not applicable.

**Data Availability Statement:** Not applicable.

**Conflicts of Interest:** The authors declare no conflict of interest.

## Appendix A. Survey Questionnaire: I Confirm That There Is a Reference to the Attachment

The importance of castles for the functioning of cultural tourism in the opinion of the inhabitants of Central Europe

The subject concerns scientific research at the University of Rzeszow and IMACSSS (International Martial Arts and Combat Sports Scientific Society). We kindly ask you to answer all the questions, choosing the most appropriate one from among the proposed ones, and to provide broader statements about your trips (question 12), during which you saw medieval castles. Thank you for your time

1. Was the trip to the castle/castles the main purpose of your trip?
   Yes
   No
2. If so, it was a trip to: country, town, ...
3. How many times have a castle or group of castles been the main purpose of your trip (please give the number)?

4. Did you participate in tourist trips, in which one of the points of the program was visiting castles?
   Yes
   No
5. If so, it was a trip to: country(ies), town, …
6. If a stay in a castle is one of the elements of a tourist trip, how can you describe it?
   uninteresting/waste of time
   moderately interesting
   good variety of stay
   very interesting
   an excellent lesson in living history
7. Please indicate the main motive for your trips to the castle(s).
   expanding historical knowledge
   interest in defensive architecture
   active or passive participation in historical reconstruction
   willingness to have an adventure
   willingness to ad variety to your stay in the tourist region
8. What additional attractions, do you think, should accompany visiting the castle?
   historical reconstructions
      staged tales about castles (stories, light-sound shows, themed shows, etc.)
      court music concerts
      craft shows
      Food and drink tasting
9. What emotions did you experience while visiting the castle?
      meditation and reflection
      admiration for the craftsmanship of old architecture
      'a thrill' associated with participation in historical shows
      fear and concern
      the feeling of the passing ages
      boredom and weariness
10. Which castles from the mentioned periods have the greatest tourist value for you?
      Medieval
      Renaissance
      Baroque
      Enlightenment
      From the 19th and 20th century
11. If your trips resulted from your interest in the traditions of military and martial arts, they concerned in particular:
      historical reconstruction
      museums and weapons shows
      museums and costume shows from old Times
      military architecture
      entertainment, curiosity
12. Which of the visited castles in Europe, according to you, is the most interesting in terms of tourism?
      Another questions concern only medieval castles. We are asking for comments about your trips, during which you saw MEDIEVAL castles. We are also asking for some general comments about such trips, their motives, goals, and impressions. We ask for a longer statement that will contain answers to the following questions:
12.1. Where did You go (countries or places), how many times?
12.2. Were these objects visited by the way, or was the trip undertaken specifically to see the castle?
12.3. In your case, was it related to your interest in the history and culture of old eras, or perhaps with the traditions of military and martial arts?

12.4. Which historical object (among castles visited or viewed from outside) made the greatest impression on you and why?

Metrics
Age:
Sex:
Male
Female
Education:
Basic
junior high school
medium
higher
Country:
Professional position:
working
not working (in productive age)
pensioner
annuitant
student
Place of living:
city
village

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
