# Peer review of "Defensive Strongholds and Fortified Castles in Poland—From the Art of Fortifications to Tourist Attractions"

_sustainability, doi:10.3390/su14063209_

Round 1

Reviewer 1 Report

The paper refers to a re-submission, then my comments will take this fact into consideration.

Apparently, the article has not been substantially changed compared to its original version: in respect to the remarks made on the first review, only two parts have been added, one referring to the selection of the interviewees’ sample and another one listing questions included in the questionnaire; but many other issues have remained unsolved:

- a basic incoherence still marks the work, as the article’s focus is tourists’ motivation and perception whereas the Literature Analysis (or, potentially, the Background) keeps on leaving wide space to the castles’ description; only a few lines have been added on those issues, unfortunately too brief;

- the use of that wide description of castles is still unclear, since this part does not affect the results’ discussion, nor does it help in indicating specific, tailor-made tourist promotion strategies;

- the inclusion of the 10 European castles in the ‘Case studies’ section is incorrect and misleading the reader;

- the whole part in Lines 339-379, with the respective charts (some of which still do not show the X-Y axes’ names; Tab.1, mentioned in Line 319, is also missing), should be moved from the ‘Methodology’ to the ‘Results’ section;

- the part in Lines 387-439 should be moved to the ‘Introduction’ or ‘Case description’, as it has nothing to do with the Results;

- the part in Lines 440-453 should be moved to a specific ‘Discussion’ section and integrated;

- Fig.6 shows that the study has concerned the European castles instead of the Polish ones, in contrast with the title;

- between the presentation of statistical results and the identification of future research areas, a fundamental phase of discussion is missing, in such a way to support and explain the conclusions the authors get to;

- the integration of new bibliographic references on specific motivational issues of tourism is appreciable, but, unfortunately, these do not give rise to reflections and discussions in the text.

In the ‘Conclusions’ section, in particular, a large part of statements (Lines 508-516) seems to be more the authors’ personal judgement rather than a direct result of interviews or of data elaboration (what is meant there by ‘trust’? There are no examples explaining this point). In the final part, the only objectively acceptable conclusion is in Lines 517-519, really too scant to motivate the study.

My general impression is that only little shallow modifications have been brought, and that the whole work remains very confused, especially in the article’s structure, as authors seem not to have full awareness of what the content of each section (‘Literature review’, ‘Background’, ‘Methodology’, ‘Results’, ‘Discussion’) should be. Ultimately, a real and adequate discussion of results against literature’s premises is missing: the results are not discussed in the light of the literature review simply because there is (almost) no Literature review on issues related to people-centred tourism and heritage perception.

Except from brief description of charts, the work is chiefly a descriptive essay on Polish and European castles and strongholds and their history and meaning, and has very little to do with tourism-related issues, that are dealt with very shortly. In Fig.7, for example, there is no in-depth analysis: who (age, sex, education level, provenience) prefers museums? Costumes? Entertainment? And why? So a structured answer to the scientific problem of the paper cannot be given.

Furthermore: What can be derived from the results that can be useful to define the most adequate promotion strategies for Polish castles? What about differences or similarities in Polish against foreign visitors’ attitudes or preferences? What about tourism policies for similar heritages of other countries? Why there is no reflection or comparison between Polish and international tourism policies? All this kind of analysis is missing. There is no insight into those issues, nor are the results scrutinized in that light.

Ultimately, it seems that the authors start from their personal conviction of the castles’ importance, and that, at the end of their survey, they have assessed the same importance among visitors simply because they were willing to fill in the questionnaires, and they consider this sole result satisfying, for which the sample’s profiling seems to be totally irrelevant and then unnecessary.

Overall, I think that the work is not suitable for publication.

Author Response

The paper refers to a re-submission, then my comments will take this fact into consideration.

Apparently, the article has not been substantially changed compared to its original version: in respect to the remarks made on the first review, only two parts have been added, one referring to the selection of the interviewees’ sample and another one listing questions included in the questionnaire; but many other issues have remained unsolved:

  • a basic incoherence still marks the work, as the article’s focus is tourists’ motivation and perception whereas the Literature Analysis (or, potentially, the Background) keeps on leaving wide space to the castles’ description; only a few lines have been added on those issues, unfortunately too brief;

My answer

You require a marketing and management perspective on tourism. But that's not the point here. The main scientific problem here is to present the essence of the phenomenon of medieval fortified castles, especially Polish ones, in relation to the concept of sustainability (which concerns the essence of this Special Issue). This article is specifically about and caring for cultural heritage in the sense of being sustainable in this regard. So it is a discussion of the cultural phenomenon of medieval castles with examples, and then - about the importance of this phenomenon for today's tourists. A fragment explaining this has been added to the Introduction - lines 137-149 (green).

  • the use of that wide description of castles is still unclear, since this part does not affect the results’ discussion, nor does it help in indicating specific, tailor-made tourist promotion strategies;

  • the inclusion of the 10 European castles in the ‘Case studies’ section is incorrect and misleading the reader;

My answer

This is the description of the cultural phenomenon in accordance with the already explicitly presented scientific problem.

  • the whole part in Lines 339-379, with the respective charts (some of which still do not show the X-Y axes’ names; Tab.1, mentioned in Line 319, is also missing), should be moved from the ‘Methodology’ to the ‘Results’ section;

My answer

The table content has been improved.

  • the part in Lines 387-439 should be moved to the ‘Introduction’ or ‘Case description’, as it has nothing to do with the Results;

My answer

This is a description of the use of Polish castles as a result of the analysis of the content of source materials and literature on the subject (analytical method, monographic method) - in accordance with the indicated cultural phenomenon that requires scientific study.

  • the part in Lines 440-453 should be moved to a specific ‘Discussion’ section and integrated;

My answer

The snippet has been renamed "Discussion".

  • 6 shows that the study has concerned the European castles instead of the Polish ones, in contrast with the title;

My answer

These are the answers to the question about other European castles (apart from Poland).

  • between the presentation of statistical results and the identification of future research areas, a fundamental phase of discussion is missing, in such a way to support and explain the conclusions the authors get to;

My answer

Additional information on the results of own research plus their interpretation has been added (lines: 423-470).

  • the integration of new bibliographic references on specific motivational issues of tourism is appreciable, but, unfortunately, these do not give rise to reflections and discussions in the

My answer

This is also a consequence of the fact that this research is not about the economic dimension of tourism, but the perception of objects.

In the ‘Conclusions’ section, in particular, a large part of statements (Lines 508-516) seems to be more the authors’ personal judgement rather than a direct result of interviews or of data elaboration (what is meant there by ‘trust’? There are no examples explaining this point). In the final part, the only objectively acceptable conclusion is in Lines 517-519, really too scant to motivate the study.

My answer

A paragraph was added (lines 603-606) with a conclusion from the statistically significant results obtained.

My general impression is that only little shallow modifications have been brought, and that the whole work remains very confused, especially in the article’s structure, as authors seem not to have full awareness of what the content of each section (‘Literature review’, ‘Background’, ‘Methodology’, ‘Results’, ‘Discussion’) should be. Ultimately, a real and adequate discussion of results against literature’s premises is missing: the results are not discussed in the light of the literature review simply because there is (almost) no Literature review on issues related to people-centred tourism and heritage perception.

My answer

In the current form, the grounding in the literature and source materials (62 items) seems sufficient. Many works are cross-referenced in Sections 1, 2, and 4.

Except from brief description of charts, the work is chiefly a descriptive essay on Polish and European castles and strongholds and their history and meaning, and has very little to do with tourism-related issues, that are dealt with very shortly. In Fig.7, for example, there is no in-depth analysis: who (age, sex, education level, provenience) prefers museums? Costumes? Entertainment? And why? So a structured answer to the scientific problem of the paper cannot be given.

My answer

The results of the relationship between the metrics and the answers to the research / questionnaire questions are now in a specially added table 1 and in the first part of the Results section.

Furthermore: What can be derived from the results that can be useful to define the most adequate promotion strategies for Polish castles? What about differences or similarities in Polish against foreign visitors’ attitudes or preferences? What about tourism policies for similar heritages of other countries? Why there is no reflection or comparison between Polish and international tourism policies? All this kind of analysis is missing. There is no insight into those issues, nor are the results scrutinized in that light.

My answer

The economic dimensions of tourism were not the focus of these studies.

Ultimately, it seems that the authors start from their personal conviction of the castles’ importance, and that, at the end of their survey, they have assessed the same importance among visitors simply because they were willing to fill in the questionnaires, and they consider this sole result satisfying, for which the sample’s profiling seems to be totally irrelevant and then unnecessary.

My answer

The subject concerned research among students of tourism and recreation, and among people interested in cultural tourism.

Overall, I think that the work is not suitable for publication. My answer

Thanks to your criticism, the manuscript as it stands is probably much better.

Reviewer 2 Report

This is a much improved version of an article that I read previously.  It is now a fascinating account of the development of Polish castles.  However, there are still some problems with the manuscript.  It would be better to focus on Polish castles as there is really a contribution to knowledge here.  I have no idea why all the other castles are listed on pp7-8 and I think that section should be deleted as the evidence collected is only for Polish castles.  EVen if visitors mentioned other castles in Europe (which were in anycase different to the ones listed on pp7-8) that is still the results of a survey in Poland. The others look like a random selection of castles that I cannot see any rationale for.  It would be much better to have a map of castles in Poland, as readers might not know where they are, than castles in Europe, many of which will be familiar  to people anyway. The methodology still needs some further explanation.  Was this a survey of visitors to Polish castles?  When was it carried out and how?  Or was it a survey of one particular castle but asking tourists about castles in general?  Why were students interviewed as suggested in the methodology but the findings suggest that there were a range of people?  The results are presented in some rather illegible graphs but not really discussed.  What is the significance of the gender/age/employment etc. status of people for cultural tourism to Polish castles?  When was the survey carried out? Which year? what time of year?  Who carried out the survey and how was the sampling done?  Finally at the beginning there is a long discussion about enthusiasts and fighting reconstructions but this is not taken up in the discussion or conclusions - how can the visits to Polish castles connect with these enthusiastic amateurs? More could be made of this as it is an important aspect of cultural tourism. The conclusions are generally rather weak so they could be improved by referencing back to these earlier discussions. 

Author Response

This is a much improved version of an article that I read previously. It is now a fascinating account of the development of Polish castles. However, there are still some problems with the manuscript. It would be better to focus on Polish castles as there is really a contribution to knowledge here. I have no idea why all the other castles are listed on pp7-8 and I think that section should be deleted as the evidence collected is only for Polish castles. EVen if visitors mentioned other castles in Europe (which were in anycase different to the ones listed on pp7-8) that is still the results of a survey in Poland. The others look like a random selection of castles that I cannot see any rationale for. It would be much better to have a map of castles in Poland, as readers might not know where they are, than castles in Europe, many of which will be familiar to people anyway. The methodology still needs some further explanation. Was this a survey of visitors to Polish castles? When was it carried out and how? Or was it a survey of one particular castle but asking tourists about castles in general?

My answer

The main scientific problem here is to present the essence of the phenomenon of medieval fortified castles, especially Polish ones but not only, in relation to the concept of sustainability (which concerns the essence of this Special Issue). This article is specifically about and caring for cultural heritage in the sense of being sustainable in this regard. So it is a discussion of the cultural phenomenon of medieval castles with examples, and then - about the importance of this phenomenon for today's tourists. A fragment explaining this has been added to the Introduction - lines 137-149 (green).

In line 237 it is stated that the choice was made by "the author of this study idea". It is indeed a subjective selection, but it reflects the specificity of both Polish castles and the entire cultural phenomenon. However, due to pandemic limitations, the questionnaire was sent by e-mail and filled in electronically.

Why were students interviewed as suggested in the methodology but the findings suggest that there were a range of people? The results are presented in some rather illegible graphs but not really discussed. What is the significance of the gender/age/employment etc. status of people for cultural tourism to Polish castles? When was the survey carried out? Which year? what time of year? Who carried out the survey and how was the sampling done?

My answer

The time of research - it was spring 2021. I am not convinced whether it should be written in the article.

Each study is based on a specific sample that should be representative. Based on this claim, efforts were made to test as many diverse groups of people as possible. Therefore, the study included a group of 614 people, marked by gender, age and education. When creating the survey, questions about the socio-demographic characteristics of the person completing the survey were taken into account. These are questions about gender, age, education, place of residence or profession. The use of these questions allows you to get to know the respondent in order to better understand the answers collected in the survey.

A more detailed description of the results has been added – lines 423-470.

Finally at the beginning there is a long discussion about enthusiasts and fighting reconstructions but this is not taken up in the discussion or conclusions - how can the visits to Polish castles connect with these enthusiastic amateurs? More could be made of this as it is an important aspect of cultural tourism. The conclusions are generally rather weak so they could be improved by referencing back to these earlier discussions.

My answer

In 4.1. the educational aspect of the use of this type of historical objects has been discussed.

New information about the results (correlations of responses with personal data) and an additional conclusion were added, taking into account the gender variable in relation to the interest in military traditions

Reviewer 3 Report

The authors have made some improvements. Some critical points regarding methodology and results are have still remained.

1) information about the field work must be added - when and how was the survey practically conducted, what are ethical considerations.

2) There is no information about the research hypothesis (es) and research questions. What are they (must be listed)? It is not clear what exactly was tested and what are the results and conclusions related to the hypothesis. What is meant by the research questions? Are they the same from the questionnaire? Normally we define general research question(s) in the qualitative study and look for the answers; in this case it would be enough with the hypothesis.

3) There are no clear answers on the hypotheses in the research results' part due to lack of clearly defined hypotheses in the introduction or methodology part.

4) Discussion is missing - how the results presented in the article differ/are similar to those of other studies referred in the article?

Author Response

The authors have made some improvements. Some critical points regarding methodology and results are have still remained.

  • information about the field work must be added - when and how was the survey practically conducted, what are ethical

My answer

The time of research - it was spring 2021. Due to pandemic limitations, the questionnaire was sent by e-mail and filled in electronically.

This type of survey (diagnostic survey method), which does not concern sensitive data, does not require the consent of the ethics committee. At the same time, each of the respondents (adults) gave consent, and the survey was anonymous.

  • There is no information about the research hypothesis (es) and research questions. What are they (must be listed)? It is not clear what exactly was tested and what are the results and conclusions related to the hypothesis. What is meant by the research questions? Are they the same from the questionnaire? Normally we define general research question(s) in the qualitative study and look for the answers; in this case it would be enough with the hypothesis.

My answer

In the Introduction (lines 137-149), an explanation of a scientific problem has been added. This problem in the empirical part, in the Methodology (336-344), is further specified in the form of research questions translated into questionnaire questions.

There is no consensus among methodologists as to whether hypotheses are to be made always or only when we are looking for a dependency problem. It is about the reception of a cultural phenomenon more qualitatively.

  • There are no clear answers on the hypotheses in the research results' part due to lack of clearly defined hypotheses in the introduction or methodology

  • Discussion is missing - how the results presented in the article differ/are similar to those of other studies referred in the article?

My answer

As I answered above, the methodological construction here is based on the deliberate omission of hypotheses.

The main problem is solved by describing the phenomenon (monographic study) and answering research questions. It was written out - Tab. 1 and lines 423-470.

Literature references are included in sections 1, 2 and 4, and a summary of considerations - in Summary and Conclusions. This last section has also been modified.

Thank you for your constructive criticism.

Round 2

Reviewer 1 Report

Overall, the article in its resubmitted version, meets only some of the issues raised while leaving others unsolved.

A deepening of literature and of existing studies was required on the themes of perceived cultural value and visitors’ motivation, that the authors consider, in their notes, as pertaining to the marketing and management field and, then, refuse to address in the work. In my opinion, heritage perception and motivation for the visit closely pertain to the topic expressed in the work’s title itself and would then deserve much more attention in the Literature review; also the questionnaire itself focuses on visit motivation (Question 3) and on the perceived tourist value (Question 5, Question 6). For these reasons, it is not clear why there is no Literature review on motivational issues and why the authors consider them pertaining to the fields of marketing and economics (i.e., “off-topic”). There is no logic in all this. Still, many studies exist on the theme, and the authors keep on ignoring them in the literature background, though appreciably widening the list of references (furthermore, heritage value perception and visitors’ motivation would offer the authors a good way to develop, in their dissertation, at least the social dimension of sustainability, which they claim to be a core concept of the paper).

In this way, the study aims to address the tourist value perceived by visitors without addressing this theme in the literature review. This is a limitation that the revision has left unsolved.

A second inconsistence lies in the authors stating that sustainability is a key concept of their work. Actually, the term “sustainability” only appears in this last version of the work (since it was totally missing in the previous one) and is not really addressed in a convincing way. It does not inform, in any of its facets (environmental, social or economic sustainability), the study, the questionnaire’s structuring and analysis, the ‘Results’ or the ‘Discussion’: there is no trace of “ecological care” (Line 142) and “sustainable tourism” (Line 143) related reasoning in any of those sections. The value underlined throughout the text and particularly in Lines 160-165 is essentially linked, beyond the architectural and construction features, to the historical victories and to glory, and no other value related to sustainability is put into light, demonstrated or discussed. Then, the claim that the castles’ sense of being sustainable is the main topic of the work is not an argument.

Consequently, the text in Lines 137-149 mentioning issues that are not subsequently addressed neither in the survey study nor in the discussion, is out of context and my strong conviction is that it should be eliminated since it does not really add, as it is, to the work; alternatively, in order to match the main theme of the journal, it should be further developed, at least in the “Discussion”.

The result discussion added in Lines 428-470 is appreciable and represents a good improvement of the work, together with the wider explanation of the methodology.

I do not think that all the issues raised through my comments have been satisfactorily addressed, but the modifications brought in the article structure and in the “Methodology”, “Results” and “Discussion” sections represent some partial improvements that make the article just barely fit for publication.

Specific comments:

Lines 70-71: “the fortified settlement protecting the settlement” (repetition of terms)

Line 80: “families of richer noble families” (repetition of terms)

Lines 91-92: verb missing

Lines 129-130: verb missing.

Author Response

Answers for Reviewers – second round

For the 1st Reviewer

Overall, the article in its resubmitted version, meets only some of the issues raised while leaving others unsolved.

My answer:

We made further corrections for the article to be approved.

A deepening of literature and of existing studies was required on the themes of perceived cultural value and visitors’ motivation, that the authors consider, in their notes, as pertaining to the marketing and management field and, then, refuse to address in the work. In my opinion, heritage perception and motivation for the visit closely pertain to the topic expressed in the work’s title itself and would then deserve much more attention in the Literature review; also the questionnaire itself focuses on visit motivation (Question 3) and on the perceived tourist value (Question 5, Question 6). For these reasons, it is not clear why there is no Literature review on motivational issues and why the authors consider them pertaining to the fields of marketing and economics (i.e., “off-topic”). There is no logic in all this. Still, many studies exist on the theme, and the authors keep on ignoring them in the literature background, though appreciably widening the list of references (furthermore, heritage value perception and visitors’ motivation would offer the authors a good way to develop, in their dissertation, at least the social dimension of sustainability, which they claim to be a core concept of the paper).

In this way, the study aims to address the tourist value perceived by visitors without addressing this theme in the literature review. This is a limitation that the revision has left unsolved.

My answer:

References to literature were introduced and added to the Discussion - lines 530-531 and 587-591.

A second inconsistence lies in the authors stating that sustainability is a key concept of their work. Actually, the term “sustainability” only appears in this last version of the work (since it was totally missing in the previous one) and is not really addressed in a convincing way. It does not inform, in any of its facets (environmental, social or economic sustainability), the study, the questionnaire’s structuring and analysis, the ‘Results’ or the ‘Discussion’: there is no trace of “ecological care” (Line 142) and “sustainable tourism” (Line 143) related reasoning in any of those sections. The value underlined throughout the text and particularly in Lines 160-165 is essentially linked, beyond the architectural and construction features, to the historical victories and to glory, and no other value related to sustainability is put into light, demonstrated or discussed. Then, the claim that the castles’ sense of being sustainable is the main topic of the work is not an argument.

My answer:

The authors wanted to justify the description of the phenomenon of medieval castles, which should be protected as a heritage requiring sustainable tourism or other forms of development.

Consequently, the text in Lines 137-149 mentioning issues that are not subsequently addressed neither in the survey study nor in the discussion, is out of context and my strong conviction is that it should be eliminated since it does not really add, as it is, to the work; alternatively, in order to match the main theme of the journal, it should be further developed, at least in the “Discussion”.

My answer:

In lines 568-571 a reference to the idea of sustainability in this type of tourism has been added.

The result discussion added in Lines 428-470 is appreciable and represents a good improvement of the work, together with the wider explanation of the methodology.

My answer:

We are very grateful for your time and valuable comments on the work. It is a great honor for us to read these words, which are a great motivation for further activities in the field of science.

I do not think that all the issues raised through my comments have been satisfactorily addressed, but the modifications brought in the article structure and in the “Methodology”, “Results” and “Discussion” sections represent some partial improvements that make the article just barely fit for publication.

My answer:

Thank you very much for your thorough analysis and valuable comments. We really appreciate your commitment and sincerity and guidance that can contribute to improving the quality and development of our further research activities.

Lines 70-71: “the fortified settlement protecting the settlement” (repetition of terms)

My answer:

It was corrected.

Line 80: “families of richer noble families” (repetition of terms)

My answer:

It was corrected.

Lines 91-92: verb missing

My answer:

It has been completed.

Lines 129-130: verb missing.

My answer:

It has been completed.

Thank you for your constructive criticism.

                                               On behalf of the authors' team, the team leader.

This manuscript is a resubmission of an earlier submission. The following is a list of the peer review reports and author responses from that submission.

Round 1

Reviewer 1 Report

 The paper intends to propose various use of medieval monuments of the art of fortifications for giving a better understanding of visitors’ evaluated value of heritage sites and corresponding behaviors.

 However, previous studies dealing with various uses and usefulness, and appropriate design and analysis of research subjects are insufficient. In particular, this study seems not have a proper format. For example, the background and purpose of the study should be clearly described in the introduction. However, the study does not even know where the end of the introduction is.

 Most importantly, there is a lack of review of previous studies in evaluating or utilizing the value of the heritage. In particular, questionnaire design should be conducted through comparative review with previous studies according to the purpose of the study.

Line 46 and 86.  Please check below sentence. Is it not a sentence but a title?

Defensive castles in the architecture and traditions of warrior cultures

Pre-Polish and Polish defensive strongholds and castles

Line 268

A logical survey design is required for the study subjects

For example, why was the age of the study subjects limited to 13 to 74 years old? In general, it is common to exclude teenagers because they have low reliability as subjects of the survey.

 In particular, 13 years old is related to children or adolescence, so they are not selected as subjects in the general visitor survey except when they are related to children or adolescence. If there is a reason, a basis for prior research or reliable data should be presented.

Furthermore, as you showed in Figure 1, the dominate ages is between 20 and 25. Then, why you select this age gap.

 In addition, when you wanted to ask about the option of the inhabitants of Central Europe, why were the respondents limited to Poland, Ukraine, and the other Europe? where does the several other Europe refer to? More sophisticated research designs are required to fit the research purpose.

Line 426

I would also recommend clarifying the implications of the study by clearly dealing with conclusions and discussions in two different paragraphs.

Reviewer 2 Report

The proposed article aims to investigate the importance of castles for cultural tourism in the opinion of Central Europe inhabitants, based on a statistical survey. Its main contribution is in the originality of the specific destination selected (fortified castles); a potential strength of the work lies in the topical character of the attention towards users’ perception in cultural tourism.

However, the article presents important criticalities. One main problem is in the structure of the work itself. The whole work is somehow unbalanced, since it combines a very long Introduction section with very brief sections dedicated to research results and discussion.

‘Introduction’ - Indeed, the ‘Introduction’ section addresses the importance of castles and fortified buildings in the culture and tradition of Poland without really delving into tourism-related issues.

Also the reason for describing other castles in Europe so extensively is rather unclear, given that the title specifically refers to Polish castles, and - apart from their connections to the art of war - there is no mention of the peculiarities of Polish castles as artefacts or tourist destinations compared to other castles in Europe or across the world.

The description of 10 castles in Europe (lines 193-267) is somehow confusing and does not let us ascertain clearly which castles are the article’s case studies. Actually, it is not clear what the use of this part is with respect to the article’s purpose. Indeed, the contribution of Polish castles to the fortified architecture of Europe, or their peculiarities in the European scene, are not even mentioned, apart from one brief quotation in the very last lines (262-267). Moreover, descriptions of those castles are rather heterogeneous: some of them are very detailed (223-231), compared to others (246-247). Then, what is their real contribution to the whole work?

In contrast with the focus of the work (cultural tourism and people’s perceptions), issues of cultural tourism and motivational investigations in the international discourse are absent from the Introduction, nor are a Literature review or a Background section to be found addressing those themes, which is an important flaw given the purpose of the work and considered the wide international debate on the topic of the article’s scientific problem.

‘Methodology’ - The ‘Methodology’ section is also very unclear, since it gives no information on the sample selection, on the content of the questionnaire and on the selected case studies. How was the sample defined? With which criteria? Were the respondents selected among Polish castles’ visitors, general tourists (then, anyway motivated towards the theme) or plain UE inhabitants? What kind of questions were asked to respondents? Were respondents interviewed about visits to specific castles or not? Did foreign and Polish inhabitants visit the same castles or not? This could be very important for discussion, as it involves differences in the value attached to tourist destinations.

Statements in lines 298-299 (“observations are evenly distributed over the intervals”) and lines 299-300 (“the dominant age range of the respondents is between the ages of 20 and 25”) seem to be quite contrasting.

It is not very clear what the use of so many descriptions of construction techniques is, whereas more details about strategies for tourist exploitation of the castles could start interesting reflections and comparisons, useful for result discussion.

Although the complete questionnaire has been correctly inserted in the Appendix, still some information about the kind of questions handed out should be given in the Methodology section. Lacking this information, it is very hard to draw from the article’s reading a clear understanding of what the survey really wants to investigate and through which indicators.

(Incidentally, does Question 12 refer to a sample of destinations or to any castle in Europe? It is not clear.)

Fig.1: incomplete chart (the names of X and Y axes are missing)

Fig.2 to 4: incomplete chart (values on X-axis and the names of X and Y axes are missing)

Fig.5: illegible chart texts.

Moreover, the insertion of the “working age” (48) cases among the categories seems rather incorrect, given that also the “working” and “learning” as well as “non-working” cases can fall within the working age (actually, the working age range should be specified).

The survey sample seems also rather unbalanced with reference to the interviewees’ countries; few (1 to 5) cases from foreign countries cannot be reasonably considered as representative as 582 cases from Poland, so there could be not much sense in combining data.

More generally, if the article’s focus is represented by Polish castles, investigating the distinction between Polish (domestic tourism) and European (outbound tourism) visitors could be important, since it can directly affect motivations and attached values.

‘Results’ - In the 'Results' section, it is not clear whether the 3.1 subsection is deduced from statistical analyses or is, rather, a description that could better integrate the Introduction section. Also the first three paragraphs of the 3.2 subsection should be better moved to the Introduction, as they have nothing to do with the survey results.

Lines 397-398: “related to tourism, tourism geography and tourism geography” (please, check or rephrase)

Fig. 7: illegible chart

Overall, this section is very obscure and the result analyses are dealt with in a very few words; the charts do not match the results of each single question but above all, a systematic discussion of the results of each question is missing, so there is very little information that can be inferred on the general topic, and almost no useful insight.

‘Summary’ - The first paragraph of the section offers information on one single building, which could be inserted, instead, in the Introduction.

Overall, the 'Summary' section does not offer specific conclusions from the results.

Overall structure

The article's title refers to Polish castles, but its content is widely related to European castles, which could potentially confuse the readers.

The abstract offers no information on the research results and conclusions, not much more than what is to be found on the respective sections.

There is no Literature review or Background section in the field of cultural tourism and of cultural tourists’ perceptions, motivations and behaviour.

A specific section for the identification and description of case studies, separated from the Introduction, could be probably useful.

A real and in-depth result analysis as well as a Discussion of results against the background premises are missing.

The Bibliographic references section is scarce in itself, outdated and focused on Poland; it totally overlooks the international debates and the whole work done in the global research sector along the last decade about cultural tourism and visitors motivations, and this is in itself a “serious flaw” of the work, which affects its overall quality.

Final comment

Ultimately, despite its interesting purpose, the article does not sensibly add to knowledge on the basic scientific problem. A wide part is dedicated to the castles’ description, but lacking in results' analysis, validation and discussion against the background and previous research. Overall, and specifically with reference to the identification of case studies and the definition of the sample, the article is confused and seriously lacking in scientific rigour and intelligibility, deserving a serious rethinking and restructuring of the work. A strong suggestion is to explore the international literature on the themes of cultural tourism and of visitors’ evaluation of destinations. A deep revision of the English language is also warmly suggested.

Reviewer 3 Report

  1. There is no setting out of a problem - for example the issue of how castles could be a form of cultural tourism.  The article goes off in several different directions
  2. The empirical analysis and presentation is poor. The graphs and tables are not informative and it is not clear what the statistical material is showing
  3. There is a selection of castles from all over the world but it is not clear how they are selected or whey
  4. The article lacks historical detail and depth

Reviewer 4 Report

The topic of the article is interesting and useful for the international readership because problems related to the cultural heritage in modern society and diversification of its changing manifestations, symbolic meaning or economic, social and environmental aspects of tourism are topical for many countries. The authors present results of their original research, the quantitative survey has aimed to explore opinions of internationally selected sample. That is the main merit of the paper. However, there is a number of arguments that must be considered and implemented to improve the overall quality of the article. 

  1. The structure of the abstract and introduction should follow the guidelines provided by the journal. It is difficult to understand the main purpose of the study; no research questions and hypothesis are presented neither in the introduction nor in the methodology part. Therefore, it is not clear what exactly was measured and analysed  in the data analysis part.
  2. I suggest to shorten the introduction and part of the information included in its current version move to the separate section related to the literature review and analysis of previous research. Please make more clear problem statement in the introduction.
  3. The methodology description is poor and must be strongly revised. The authors state that they study "opinion of the inhabitants of Central Europe" (line 270). However, from the sample description is clear that few respondents represent also Western European countries. They must be either excluded from the research or emphasis on Central Europe must be reconsidered. Information about the field work must be added (where, how, when the survey was conducted, what was the sampling procedure, why and how children were involved in the survey, research ethics, informed consent of the respondents, etc.). The sentence "The main method of research is a diagnostic 271 survey carried out with the use of a survey on a group of N important according to the 272 statistics in the number of n = 617 respondents" is not clear at all. Wat is diagnostic survey? Maybe some reference can be added to explain this. Information about the research questions/hypothesis must be added as well as what statistical correlations/analysis was used to approve or reject hypothesis.
  4. I see no reason to include so many figures about the research sample. Even more, their quality is low, e.g. the text is not clear (Fig.5). I would suggest to focus more on data analysis. From the questionnaire template it is clear that the authors have gathered much more data and information than they actually present in the article. As there are no research questions/hypothesis defined in the paper, it is not very clear what opinions particularly were analysed, what was meant to be approved by the statistical analysis. This should be improved and added. Also sample demographic data can be well used to analyse opinions of different respondents groups (age, gender, etc.).
  5. In the discussion part, the authors should focus more on comparison of their own research results with other references and to give some suggestion related to their research questions/hypothesis.